# Artificial intelligence that determines the clinical significance of capsule endoscopy images can increase the efficiency of reading

Junseok Park[1], Youngbae Hwang[2], Ji Hyung Nam[3], Dong Jun Oh[3], Ki Bae Kim[4], Hyun Joo Song[5], Su Hwan Kim[6], Sun Hyung Kang[7], Min Kyu Jung[8], Yun Jeong Lim[3]*

1 Department of Internal Medicine, Digestive Disease Center, Institute for Digestive Research, Soonchunhyang University College of Medicine, Seoul, Republic of Korea, 2 Department of Electronics Engineering, Chungbuk National University, Cheongju, Republic of Korea, 3 Division of Gastroenterology, Department of Internal Medicine, Dongguk University Ilsan Hospital, Dongguk University College of Medicine, Goyang, Republic of Korea, 4 Department of Internal Medicine, Chungbuk National University College of Medicine, Cheongju, Republic of Korea, 5 Department of Internal Medicine, Jeju National University School of Medicine, Jeju, Republic of Korea, 6 Department of Internal Medicine, Seoul Metropolitan Government Seoul National University Boramae Medical Center, Seoul, Republic of Korea, 7 Division of Gastroenterology and Hepatology, Department of Internal Medicine, Chungnam National University School of Medicine, Daejeon, Republic of Korea, 8 Division of Gastroenterology and Hepatology, Department of Internal Medicine, Kyungpook National University Hospital, Daegu, Republic of Korea

☯ These authors contributed equally to this work.
* drlimyj@gmail.com

**Data Availability Statement:** Data cannot be shared publicly because of ethical restrictions. Capsule endoscopy data of this research, including images and results, are anonymous and non-

## Abstract

Artificial intelligence (AI), which has demonstrated outstanding achievements in image recognition, can be useful for the tedious capsule endoscopy (CE) reading. We aimed to develop a practical AI-based method that can identify various types of lesions and tried to evaluate the effectiveness of the method under clinical settings. A total of 203,244 CE images were collected from multiple centers selected considering the regional distribution. The AI based on the Inception-Resnet-V2 model was trained with images that were classified into two categories according to their clinical significance. The performance of AI was evaluated with a comparative test involving two groups of reviewers with different experiences. The AI summarized 67,008 (31.89%) images with a probability of more than 0.8 for containing lesions in 210,100 frames of 20 selected CE videos. Using the AI-assisted reading model, reviewers in both the groups exhibited increased lesion detection rates compared to those achieved using the conventional reading model (experts; 34.3%–73.0%; p = 0.029, trainees; 24.7%–53.1%; p = 0.029). The improved result for trainees was comparable to that for the experts (p = 0.057). Further, the AI-assisted reading model significantly shortened the reading time for trainees (1621.0–746.8 min; p = 0.029). Thus, we have developed an AI-assisted reading model that can detect various lesions and can successfully summarize CE images according to clinical significance. The assistance rendered by AI can increase the lesion detection rates of reviewers. Especially, trainees could improve their efficiency of reading as a result of reduced reading time using the AI-assisted model.

personally identifiable, but sensitive human study participant data. Data are available from the Institutional Data Access / Ethics Committee of Dongguk University Ilsan Hospital (contact via irb@dumc.or.kr) for researchers who meet the criteria for access to confidential data.

**Funding:** This study was supported by a grant from the Korean Health Technology R & D project through the Korean Health Industry Development Institute (KHIDI, https://www.khidi.or.kr/eps), funded by the Ministry of Health & Welfare, Republic of Korea (Grant Number: HI19C0665). The corresponding author, Dr. LYJ, received the fund. The funders did not play a direct role in writing this article, but this study could be constructed on the fund.

**Competing interests:** The authors have declared that no competing interests exist.

## Introduction

Deep learning-based artificial intelligence (AI) has demonstrated outstanding levels of achievement in image recognition [1]. The diagnostic yield of medical imaging tests relies on lesion detection, for which the use of AI we can be take the advantageous of AI [2–4]. Capsule endoscopy (CE) creates a large number of images while examining the mucosal surface of small bowel [5]. The manufacturers of capsule endoscopes develop specially designed software to increase the diagnostic yield of CE. The reviewers detect lesions by looking at the context of the images reconstructed by the software according to the image-acquisition time. The tedious inspection requires mental concentration of reviewers that can be easily compromised by the inherent limitations of human capabilities [6]. To alleviate the burden on reviewers, many studies have attempted to apply AI to CE reading and showed impressive results in detecting small bowel lesions [7–9]. However, the clinical application of AI must be preceded by practical considerations and algorithm optimization. Multiple lesions may exist concurrently in a CE image and the types of lesions can vary. Normal obstacles, including bile, bubbles, and debris may interfere with accurate examination of the small bowel. The AI must be able to handle all these findings. In addition, the effectiveness of AI should be evaluated in situations similar to those of conventional reading methods.

In the present study, we determined the clinical usefulness of a deep learning-based AI model for CE reading. The AI was developed to recognize the various types of small bowel lesions. We also conducted tests to evaluate the performance of observers using a specially designed time-based method comparing the conventional reading model and the AI-assisted reading model. We tested whether the AI-assisted reading model could shorten the reading time and increase the lesion detection rate in two groups with different levels of experience.

## Materials and methods

### Collecting image datasets for AI training

We retrospectively collected 139 CE videos from seven university hospitals, selected considering the regional distribution, with the approval of the individual institutional review boards (Dongguk University Ilsan Hospital; IRB No. 2018-10-009-002). The cases were performed between 2016 and 2019 using PillCam SB3 (Medtronic, Minneapolis, MN, USA). Using a dedicated software, Rapid Reader ver. 8.3., a total of 203,244 small bowel images were extracted from the videos in 512 × 512-pixel PNG format. After data anonymization, the small bowel images were reviewed by five gastroenterologists proficient in CE reading and were arranged according to clinical significance. Based on the consensus of at least two reviewers, the images were categorized into two classes. Images with inflamed mucosa, atypical vascularity, or bleeding were classified as significant. Images of normal mucosa with bile, bubbles, or debris were classified as insignificant (Fig 1).

### Development of CNN-based AI for reading support

Among the 203,244 images collected, we randomly selected the dataset of 200,000 images comprising 50% of significant and insignificant classes. We then separated the dataset into training images (60%), validation images (20%), and test images (20%). To train an AI for CE reading, we used a recently developed effective model, Inception-Resnet-V2 in TensorFlow-Slim (TF-Slim) library, which combines the advantages of efficient multi-level feature extraction of an inception module and deeper layers of a Resnet module [10, 11]. We used the transfer learning starting from the pre-trained weights using the ImageNet dataset provided in TF-Slim, because the dataset includes all the scene variations captured in the natural world

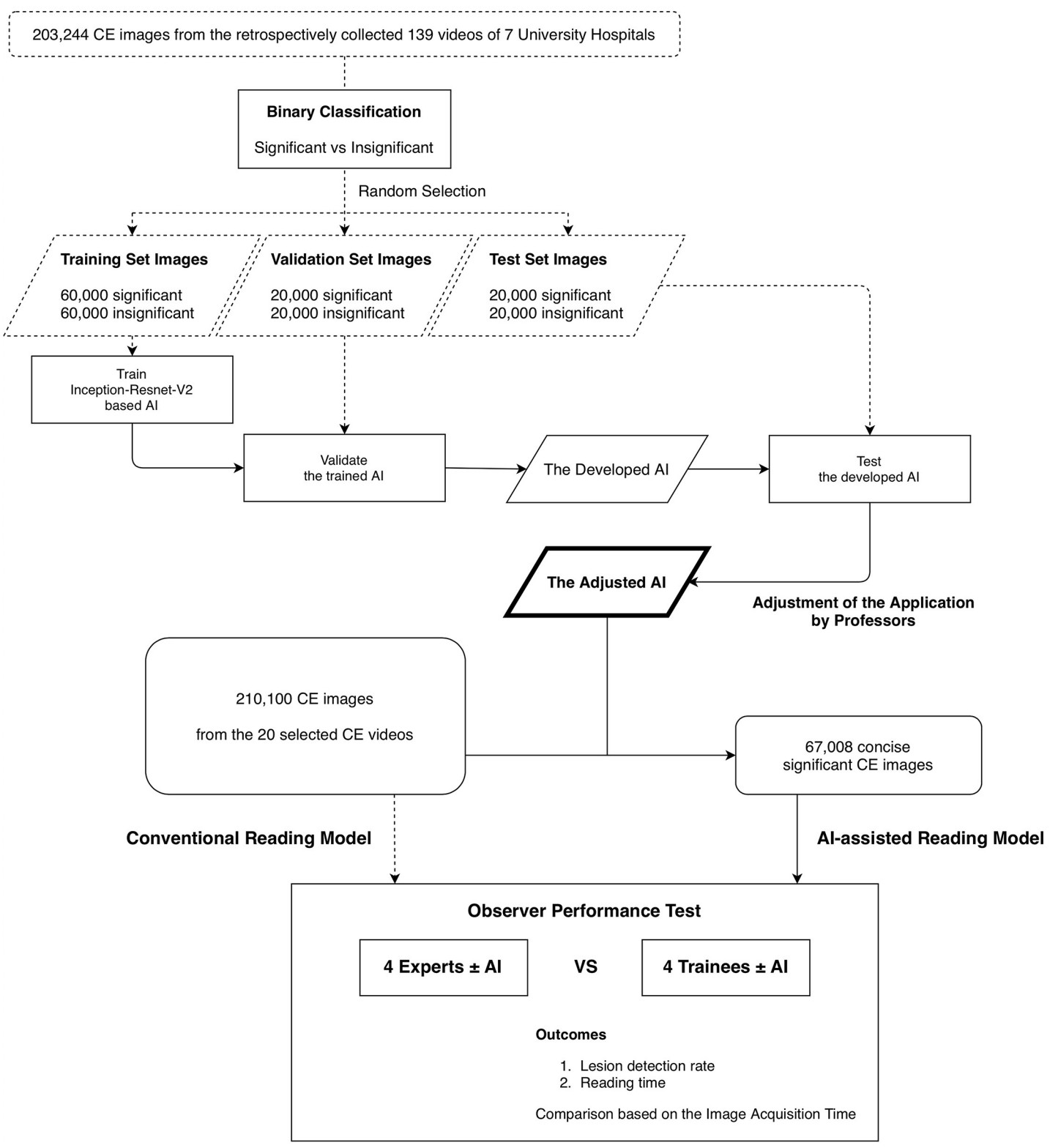

**Fig 1. Flowchart of the study design.**

**Table 1. Hyperparameters for the training of the AI.**

| Parameter | Training the only last layer from pre-trained dataset from ImageNet | Training all the layers from the trained data of the last layer |
|---|---|---|
| Batch size | 24 | 24 |
| Learning rate | 0.01 | 0.0001 |
| Epoch | 2 | 50 |
| Optimizer | RMSProp | RMSProp |

[12]. By training only the final layer from the pre-trained model, we achieved a validation accuracy of 80.29% with 2 epochs and a 0.01 learning rate (Table 1). The input image size of the deep learning network was 299 × 299, and the batch size was 24. We then achieved a validation accuracy of 98.46% by training all the layers with 50 epochs and a 0.0001 learning rate. We selected the optimal parameters to show the best validation accuracy. The final accuracy of the trained parameters to the test set was 98.34%. The elapsed time to transform image data into TFrecord (binary data rapidly readable in TensorFlow) for 40,000 images was 337.46 s. The processing time for evaluation of the transformed data was 212.74 s. Therefore, our AI enabled the processing of 71.38 frames per second. To analyze the importance of specific regions that contribute to the final class, we drew class activation maps based on channel-wise aggregation. After the final convolution layer, we applied global average pooling for each channel, corresponding to pixel-wise predicted values for the class. Using the class activation map, in which the image regions corresponding to clinical significance were indicated in red, we expected that the predictions based on the trained deep learning network would be similar to those of endoscopists (Fig 2). This map provides an insight into the role of AI-assisted reading model in decisions underlying the binary classification.

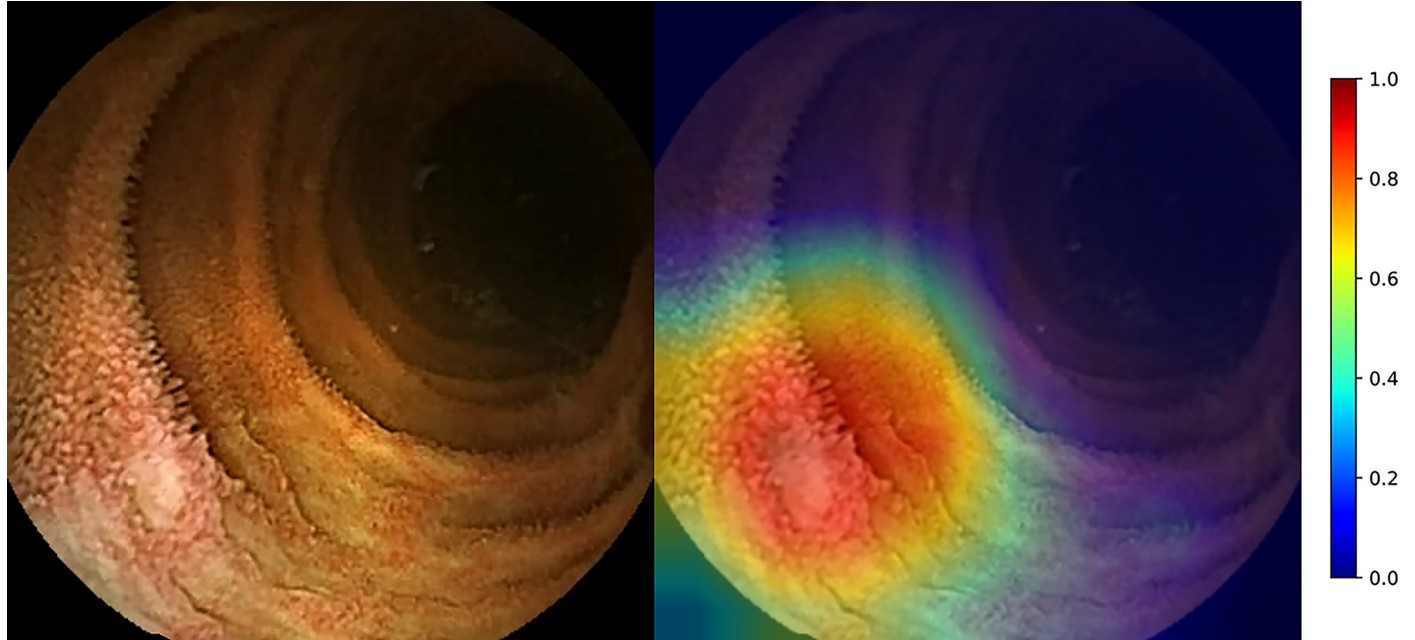

**Fig 2. Class activation map of inflamed mucosa.** Jet color map showing the normalized prediction where reddish and bluish colors are close to 1 and 0, respectively (right bar). An ulcer in the left, lower corner of the image is highlighted in red. This figure demonstrates the approximate mechanism of AI.

The calculated probabilities of significance and the results of class activation map of all the 40,000 test set images were reviewed by three professors who contributed to the classification of the AI training set. They determined that the level of AI was appropriate for assisting CE readings and agreed to set the AI threshold at 0.8 by examining the concordance between the manual classification and the calculated probability values of images (Fig 3). This allowed the AI to present an image as significant when the calculated value was over 0.8. At this cut-off value, the sensitivity, specificity, positive-predictive value, negative-predictive value, and accuracy of the AI were 96.76%, 99.46%, 99.44%, 96.85%, and 98.11%, respectively (Fig 4).

## Image acquisition time-based comparison of conventional and AI-assisted reading models

An additional 20 cases of CE that were not used for AI training were obtained considering typical clinical indications and were reviewed by three gastroenterology professors [13, 14]. The reviewers used the Rapid Reader software to capture the frames containing the clinically significant findings. The most representative image was captured when the lesion appeared over multiple frames. Reference data of lesions were constructed based on the times of image acquisition with a consensus between at least two of the reviewers. Eight other endoscopists were educated on the method of comparing lesion detection based on image acquisition time and participated in this study. They assessed the CE cases without any prior clinical information. Four (A, B, C, and D) of them were experts with more than 100 CE reading experiences. Others (a, b, c, and d) were trainees who recently learned the mechanics and software of CE through 10 reading experiences under the supervision of experts. Conventional readings were based on the Rapid Reader software. Each participant was allowed to choose the most

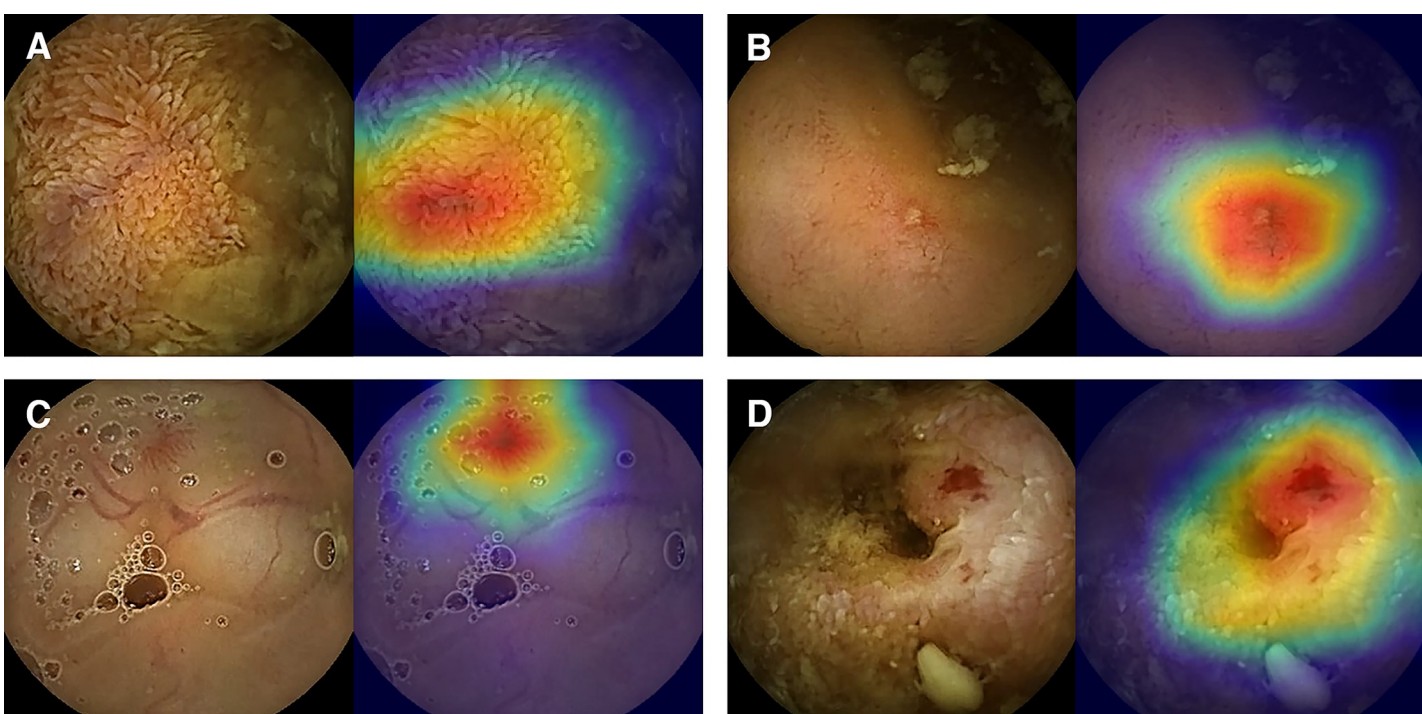

**Fig 3. Concordance of significant images between the AI and experts.** All the four images were classified as significant with 0.8 or higher probability and based on manual classification by experts (left side of the images). As the color spectrum of class activation map turns red, lesions display higher probabilities. AI can distinguish multiple findings that coexist in an image (right side of the images; A, swollen villi from debris; B, Small mucosal defect from the nearby debris; C, Vascular tuft adjacent to vessels; D, Vascular tuft surrounded by inflamed mucosa).

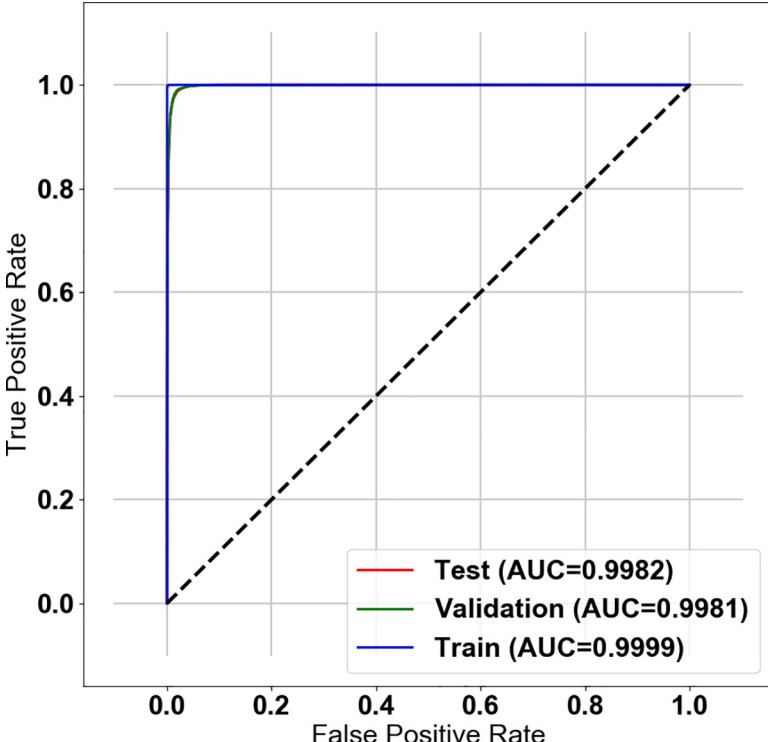

**Fig 4. Receiver operating characteristic curve of the AI for binary classification.** The receiver operating characteristic (ROC) curve of AI for detection of significant images: Area under the curves (AUCs) were 0.9982, 0.9981, and 0.9999 for test, validation, and training set images, respectively. ROC curve and AUCs shown that the training model is well fitted to all of training images as well as there is little degradation of validation and testing performance from the training model.

comfortable mode of review. However, the reading pace did not exceed ten frames per second (FPS) of quad view. Similar to the method used to generate the reference findings, a single representative frame was reported for a lesion. The AI-assisted reading model used the file explorer program of Microsoft Windows 10. To minimize the interaction between the two reading methods, the AI-assisted reading was performed at least 7 days apart from the conventional reading. Images with a probability above the threshold of AI performance test were provided to the reviewers. The images were observed under the "Extra-large icon" view mode of the file explorer. The reviewers moved all images that were considered truly significant to a separate folder. They reported the case the images belonged to and the time when they were taken. The review time in each case was recorded in minutes. Both readings were obtained on a 19-inch LCD monitor with a 1280 × 1024 resolution. To overcome the difference between the two reading models, lesion detection was compared according to the image acquisition times. If a participant reported a related lesion within a 10 s margin before and after the time at which the reference result was obtained, the lesion was considered to be detected. The lesion detection rates were evaluated per lesion. Although the participants reported several significant images within the time frame, they were considered as single lesions if the reference lesions were the same.

## Outcomes and statistical analysis

The primary outcome was the performance of AI in summarizing images of the selected CE cases using area under the receiver operating characteristic (ROC) curve to determine the

clinical significance. The extent of image reduction was calculated by removing the image with a probability below the threshold. The secondary outcome was the improvement in lesion detection rates of reviewers when using the AI-assisted reading model. The percentage of lesions detected in the references and the reading duration in each model were compared.

The correlation of continuous variables was evaluated using the bivariate correlation analysis and the quantitative difference between the subgroups was analyzed using the Mann–Whitney test. Statistical significance was set at $P < 0.05$. The Statistical Package for Social Science (version 26.0; SPSS Inc., Chicago, IL, United States) was used for statistical analysis.

## Results

### Summary performance of AI

A total of 860 lesions in the 20 CE videos were reported as references by the three professors who contributed to the classification of the AI training set. The cases included patients with representative small bowel disease and two patients with non-specific enteritis (Table 2). The small bowel transit times (SBTTs) of capsule endoscope were calculated between the acquisition times of first duodenal image and the first cecal image, and ranged between 2,751 and 30,457 s. The number of extracted images from the videos ranged between 1,947 and 34,600. Overall, 67,008 (31.9%) images were deemed significant for AI and pertained to 702 (81.6%) reference lesions. As the total number of extracted images increased, the number of images exceeding the significance threshold tended to increase (Pearson Correlation Coefficient = 0.878, $P < 0.001$). There was no statistically significant correlation among the other values (reference lesion count–total extracted images; $P = 0.278$, reference lesion count–images with significance possibility above threshold; $P = 0.817$, reference lesions count–SBTT; $P = 0.261$, total

**Table 2. Clinical characteristics of cases and summary result of AI.**

| Case No. | Diagnosis | Total extracted images | Images with significance possibility above threshold | Small bowel transit time of capsule (sec) | Reference lesion count |
|---|---|---|---|---|---|
| 1 | Bleeding on unspecified origin of jejunum | 16,500 | 7,917 (48.0%) | 28,589 | 28 |
| 2 | Crohn's disease | 13,400 | 2,013 (15.0%) | 15,998 | 56 |
| 3 | Small bowel bleeding on unspecified origin | 34,600 | 33,355 (96.4%) | 30,457 | 30 |
| 4 | Jejunal lipoma bleeding | 3,300 | 418 (12.7%) | 23,307 | 8 |
| 5 | Small bowel angioectasia | 9,800 | 1,071 (10.9%) | 17,155 | 34 |
| 6 | NSAIDs-induced enteropathy | 13,585 | 3,819 (28.1%) | 9,777 | 109 |
| 7 | Portal hypertensive enteropathy | 13,556 | 2,591 (19.1%) | 2,751 | 58 |
| 8 | Crohn's disease | 6,229 | 3,042 (48.8%) | 21,039 | 40 |
| 9 | Nonspecific enteritis | 1,947 | 98 (5.0%) | 20,988 | 6 |
| 10 | Crohn's disease | 8,755 | 239 (2.7%) | 13,343 | 14 |
| 11 | Small bowel angioectasia | 12,015 | 1,747 (14.5%) | 17,370 | 12 |
| 12 | Jejunal polyp | 8,508 | 190 (2.2%) | 13,364 | 23 |
| 13 | Small bowel angioectasia | 3,245 | 648 (20.0%) | 12,278 | 34 |
| 14 | Crohn's disease | 14,682 | 1,164 (7.9%) | 10,834 | 47 |
| 15 | Nonspecific enteritis | 6,578 | 125 (1.9%) | 29,124 | 7 |
| 16 | Crohn's disease | 4,100 | 73 (9.1%) | 25,597 | 34 |
| 17 | Crohn's disease | 14,712 | 3,807 (25.9%) | 22,433 | 172 |
| 18 | Crohn's disease | 9,642 | 3,927 (40.7%) | 7,449 | 86 |
| 19 | Crohn's disease | 7,807 | 270 (3.5%) | 25,274 | 43 |
| 20 | Crohn's disease | 7,139 | 194 (2.7%) | 19,776 | 19 |

**Table 3. Comparison of the lesion detection rates and reading times of reviewers between the reading models.**

| Reviewer | | Conventional reading model | | AI-assisted reading model | |
|---|---|---|---|---|---|
| | | Overall lesion detection count | Overall reading time (min) | Overall lesion detection count | Overall reading time (min) |
| Expert | A | 307 (35.7%) | 1,062 | 660 (76.7%) | 798 |
| | B | 278 (32.3%) | 1,140 | 644 (74.9%) | 1,116 |
| | C | 313 (36.4%) | 916 | 537 (62.4%) | 363 |
| | D | 283 (32.9%) | 1,149 | 671 (78%) | 710 |
| Trainee | a | 157 (18.3%) | 1,051 | 509 (59.2%) | 279 |
| | b | 194 (22.6%) | 2,200 | 358 (41.6%) | 627 |
| | c | 265 (30.8%) | 2,175 | 378 (44%) | 696 |
| | d | 234 (27.2%) | 1,058 | 583 (67.8%) | 747 |

extracted images–SBTT; P = 0.548, images with significance possibility above threshold–SBTT; P = 0.124). The AI suggested large numbers of images, particularly in cases associated with bleeding episodes.

## Comparison between conventional and AI-assisted reading models

With the assistance of AI, all the reviewers could find more lesions in a shorter reading time (Table 3). Four participants who were considered experts identified 295.3 (34.3%) lesions in 20 videos in 1,066.8 min on average. The four trainees reported 212.5 (24.7%) lesions in 1,621.0 min on average (Table 4). The AI suggested 67,008 images as significant for the eight participants. On average, the four experts reported 628.0 (73.0%) lesions. The mean overall reading time of the experts was 746.8 min Four trainees found 457.0 (53.1%) lesions in 587.3 min on average. The improvement of all reviewers showed statistical significance (mean lesion detection rate; 29.5%–63.1%; P = 0.01, mean reading time; 1343.9–667 min; P = 0.03).

The lesion detection rate of the experts using the conventional reading models was significantly higher than that of the trainees (experts; 34.3%, trainees; 24.7%, P = 0.029) (Fig 5). The reading time of the trainees varied widely and was not significantly different from the reading time of the experts. Both experts and trainees showed improvements in the lesion detection rates using the AI-assisted reading model (experts; 34.3%–73.0%; P = 0.029, trainees; 24.7%–53.1%; P = 0.029). The lesion detection rates for the trainees were improved to the level where they were not different from those of the experts (experts; 73.0%, trainees; 53.1%; P = 0.057). The reading time of trainees was significantly shortened (1621.0–587.3 min, P = 0.029), and was not different from that of the experts (experts; 746.8 min, trainees; 587.3 min, P = 0.343).

## Discussion

Detection of abnormal findings different from those of normal mucosa is crucial in the reading of CE images [15]. In this study, we achieved remarkable results via the deep learning-based

**Table 4. Comparison of the lesion detection rates and reading times of reviewer groups between the reading models.**

| | Conventional Reading Model | | AI-assisted Reading Model | |
|---|---|---|---|---|
| | Mean lesion detection count | Mean reading time (min) | Mean lesion detection count | Mean reading time (min) |
| Expert | 295.3±17.3 (34.3±2.0%) | 1066.8±107.8 | 628±61.7 (73.0±7.2%) | 746.8±309.6 |
| Trainee | 212.5±47 (24.7±5.5%) | 1621±654.2 | 457±107.4 (53.1±12.5%) | 587.3±211.3 |
| All | 235.9±55.1 (29.5±6.4%) | 1343.9±525.5 | 542.5±122.2 (63.1±14.2%) | 667±259.8 |

All values are presented as Mean±SD

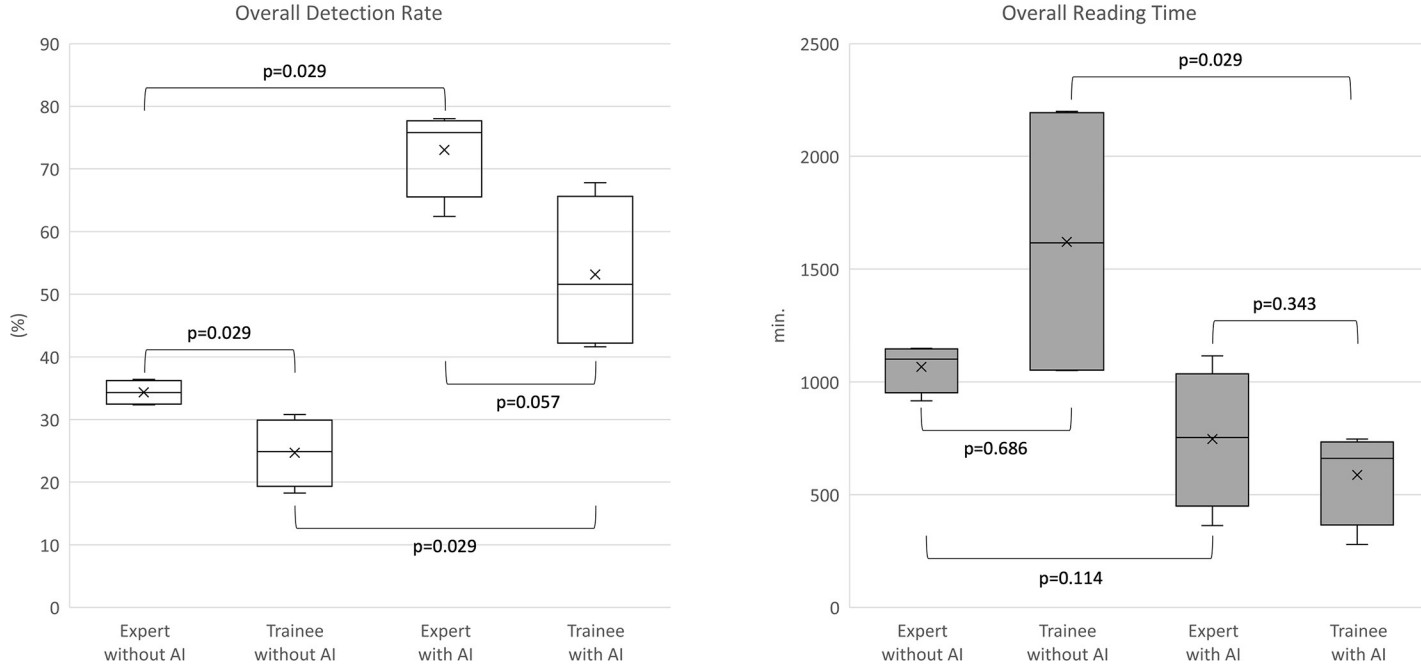

**Fig 5. Improvement in the capsule endoscopy reading using the AI-assisted reading model.**

AI to improve the lesion detection rates of reviewers. The AI was trained on a series of images classified into two categories according to their clinical significance. Terms such as "normal" and "abnormal" were not used because normal findings that were not pathological could also interfere with the observation of the small bowel mucosa. For example, "bile," "bubble," and "debris" are classified as insignificant findings. Pathological findings, including inflammation and atypical vascularity, were considered significant. Bleeding was also classified as significant. These findings, identified concurrently in a single image and in lesions of various degrees, ranging from erythema and edema to erosions and ulcers, can be recognized simultaneously. The AI must be developed to handle all these findings. Previous studies only assessed certain representative lesions with clearly distinct boundaries from the surrounding normal mucosa [7, 9]. Although there have been attempts to train an AI to recognize various types of lesions, the factors that veil lesions in the actual review of CE were underestimated [16]. AI-based image recognition has developed from earlier stages of object detection with image-level classification using AlexNet to semantic segmentation that classifies a significant region at a pixel level [17–20]. However, this detailed detection or classification cannot distinguish all the lesions in CE, and the collection of sufficient databases for each lesion remains a challenge. Furthermore, removing or adding a class in a database requires considerable effort. The AI used in this study was trained with categorized images using comprehensive binary classification. Because the category was specified to images, not lesions, the findings can coexist and do not need to be clearly separated in the same category for training. We used the approach to prepare learning materials for the AI. All the images were categorized based on the consensus of the experts. The meticulous classification empowered the AI to have high accuracy for detecting the various types of lesions in CE. The AI successfully summarized the CE images according to clinical significance. It was also demonstrated that, even if the AI provides only approximate information, it helps trainees improve their reading efficiency. The four trainees showed the statistically improved lesion detection rate (24.7%–53.1%; P = 0.029) in the

shortened review time (1621.0–587.3 min, P = 0.029). The improved results of trainees were comparable to those of experts (lesion detection rate, P = 0.057; reading time, P = 0.343).

Manufacturers of capsule endoscopes have been continuously upgrading their review software to improve the quality of reading. Nevertheless, these are managed as intellectual property. As a result, newly developed technologies are difficult to immediately integrate with the software. To evaluate the usefulness of AI as a reading support tool, objective comparison with the conventional reading model using dedicated software needs to be performed in situations that closely mimic the actual review of CE images. Because the capsule moves passively under the bowel movement and takes 2–3 pictures per second, multiple images can be obtained from the same location. Moreover, its movement can also be back and forth, and images of lesions can be obtained out of order. In the conventional reading model, reviewers report representative images that best reflect the clinical context of CE instead of analyzing each frame of images. The reviewers can detect a lesion that was missed in one image but was identified in another image. By contrast, our AI analyzes the images frame-by-frame. The number of significant images classified by the AI does not indicate the number of lesions; multiple significant images may result from a single lesion, and if there are multiple lesions in a single image, it is inevitable to classify them as one. When analyzing the 20 cases enrolled in this study a larger number of pictures were taken in the case where definite bleeding features were observed. It is known that small bowel transit time may be shortened in patients with intestinal bleeding [21]. This means that the capsule moves faster in those patients. If the effect of the adaptive frame rate of SB3 is added to this, a larger number of images can be taken in unit time. This may cause the positive correlation between the images extracted from the whole small bowel and images classified as clinically significant (Pearson Correlation Coefficient = 0.878, P < 0.001). However, the greater number of significant images does not mean that there are more lesions. Most of the bloodstain images are taken at a distance from the lesion. Images where blood spots are found can guide reviewers to find the source of bleeding. Therefore, we compared the results based on the acquisition time of reference lesions in the conventional reading model. We analyzed the ability of the AI to detect a lesion over a 20-second period, including the images obtained 10 seconds before and after the reference lesion detected using the conventional reading model. This method allows for the comparison of AI-assisted reading model at a level similar to that of the conventional reading model. Nevertheless, the lesion detection rates could be underestimated. The overall lesion detection rate of AI was 81.6% in the time-based comparison, which was less than the results of AI in other studies.

It is necessary to overcome a few limitations to utilize the AI in a clinical setting. Because the AI technology used was not specifically designed for CE and the training material comprises retrospectively collected images, it is difficult to accurately assess all types of lesions in the small bowel. Although the AI was capable of detecting various types of lesions, only the lesions that are commonly observed in the training material could be successfully detected, and its internal process to determine the significance of these lesions cannot be easily deduced. More specialized AI should be developed for the recognition of small bowel lesions, and more images of rare findings should be added to the database. Setting thresholds of AI to maximize its sensitivity increases the number of summarized images proposed as significant. This means that reviewers should make more effort while reading CE images. Prior to its application in the reading, the threshold levels need to be adjusted to practical levels. In this study, experts had the calculated significance results and the colorized class activation maps of AI on the validation set of image databases to estimate the internal process of the AI. As a result, experts could set realistic thresholds for AI to review selected CE videos. Although this could improve the reviewers' CE readings, further evaluation should be performed using a larger number of cases for a more objective evaluation.

## Conclusions

We developed an AI that was trained using carefully categorized images with a comprehensive binary classification. This AI recognized various lesions, and its utility as an assistant tool to read CE was comparatively analyzed under settings that mimic actual situations of conventional reading. The AI effectively suggested images that were summarized according to the clinical significance and improved the lesion detection rates of the reviewers (mean; 29.5%–63.1%; $P = 0.01$). More specifically, the group of trainees exhibited reduced reading times and lesion detection rates comparable to those of the group of experts (statistical difference between the two groups: lesion detection rate, $P = 0.057$; reading time, $P = 0.343$).

## Author Contributions

**Conceptualization:** Junseok Park, Youngbae Hwang, Yun Jeong Lim.

**Data curation:** Junseok Park, Ji Hyung Nam, Dong Jun Oh, Ki Bae Kim, Hyun Joo Song, Su Hwan Kim, Sun Hyung Kang, Min Kyu Jung, Yun Jeong Lim.

**Formal analysis:** Junseok Park, Youngbae Hwang.

**Funding acquisition:** Yun Jeong Lim.

**Methodology:** Junseok Park, Youngbae Hwang.

**Project administration:** Junseok Park, Yun Jeong Lim.

**Resources:** Ji Hyung Nam, Dong Jun Oh, Ki Bae Kim, Hyun Joo Song, Su Hwan Kim, Sun Hyung Kang, Min Kyu Jung.

**Software:** Youngbae Hwang.

**Supervision:** Yun Jeong Lim.

**Validation:** Ki Bae Kim.

**Writing – original draft:** Junseok Park, Youngbae Hwang.

**Writing – review & editing:** Yun Jeong Lim.

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
