## [Decision Letter · Decision Letter 0]

26 Jun 2020

PONE-D-20-15923

Artificial intelligence based on binary classification according to clinical significance of images improves the reading efficiency of capsule endoscopy

PLOS ONE

Dear Dr. LIM,

Thank you for submitting your manuscript to PLOS ONE. After careful consideration, we feel that it has merit but does not fully meet PLOS ONE’s publication criteria as it currently stands. Therefore, we invite you to submit a revised version of the manuscript that addresses the points raised during the review process.

Although, authors written a good paper, but referees are asking major revision. So, I invite you to submit a revised manuscript.

We look forward to receiving your revised manuscript.

Kind regards,

Dilbag Singh

Academic Editor

PLOS ONE

Journal Requirements:

2. Please amend either the abstract on the online submission form (via Edit Submission) or the abstract in the manuscript so that they are identical.

3.

We note that Figure 2-3 in your submission contain satellite images which may be copyrighted. All PLOS content is published under the Creative Commons Attribution License (CC BY 4.0), which means that the manuscript, images, and Supporting Information files will be freely available online, and any third party is permitted to access, download, copy, distribute, and use these materials in any way, even commercially, with proper attribution. For these reasons, we cannot publish previously copyrighted maps or satellite images created using proprietary data, such as Google software (Google Maps, Street View, and Earth). For more information, see our copyright guidelines: http://journals.plos.org/plosone/s/licenses-and-copyright.

You may seek permission from the original copyright holder of Figure 2-3 to publish the content specifically under the CC BY 4.0 license. 

If you are unable to obtain permission from the original copyright holder to publish these figures under the CC BY 4.0 license or if the copyright holder’s requirements are incompatible with the CC BY 4.0 license, please either i) remove the figure or ii) supply a replacement figure that complies with the CC BY 4.0 license. Please check copyright information on all replacement figures and update the figure caption with source information. If applicable, please specify in the figure caption text when a figure is similar but not identical to the original image and is therefore for illustrative purposes only.

Reviewers' comments:

Reviewer's Responses to Questions

**Comments to the Author**

1. Is the manuscript technically sound, and do the data support the conclusions?

Reviewer #1: Yes

Reviewer #2: Yes

Reviewer #3: Partly

2. Has the statistical analysis been performed appropriately and rigorously? 

Reviewer #1: Yes

Reviewer #2: Yes

Reviewer #3: No

3. Have the authors made all data underlying the findings in their manuscript fully available?

Reviewer #1: Yes

Reviewer #2: Yes

Reviewer #3: No

4. Is the manuscript presented in an intelligible fashion and written in standard English?

Reviewer #1: Yes

Reviewer #2: Yes

Reviewer #3: Yes

5. Review Comments to the Author

Reviewer #1: After initial screening I have observed following issues:-

1. Authors should add more comparative analyses, 2. It is advised to explain the equations present in the paper in more efficient manner. 3. Some confusing and long lines are there, authors should adjust these lines. 4. The literature can be improved by adding some recent plos one papers and also: Efficient Prediction of Drug-drug interaction using Deep Learning Models, Imputing missing values using cumulative linear regression, Visibility improvement and mass segmentation of mammogram images using quantile separated histogram equalisation with local contrast enhancement, Classification of COVID-19 patients from chest CT images using multi-objective differential evolution--based convolutional neural networks, Deep learning approach for microarray cancer data classification, Deep Transfer Learning based Classification Model for COVID-19 Disease, 5. Conclusion can be improved further by presenting more statistical computed values.

Reviewer #2: 1. Author should reconsider the title. It is very confusing.

2. Abstract is too long. Authors need to be rewrite it for better understanding.

3. It is advised that ROC curve that is Figure 4 must be reevaluated by considering ROC analysis of competitive techniques.

4. Figure 5 is confusing, so authors should provide more description about this figure.

5. It is advised to compare with more state-of-art techniques for comparative analyses.

6. Parameter setting of the proposed model should be presented in tabular form.

7. There are some typos and grammatical mistakes present in the manuscript.

8. Paper organization is missing at the end of introduction

9. Headings and sub-headings have not been marked

10. It is advisable to authors consider following useful references

Fusion of medical images using deep belief networks, Classification of COVID-19 patients from chest CT images using multi-objective differential evolution–based convolutional neural networks, Efficient prediction of drug–drug interaction using deep learning models, Deep Transfer Learning based Classification Model for COVID-19 Disease , Multi-objective differential evolution based random forest for e-health applications

Reviewer #3: Paper can be improved further

1. Add suitable research motivation

2. Add more relevant references from plos one.

3. Equations should be explained carefully

4. More comparative analysis are required

5. Add statistical analysis in conclusion or discussion session

6. PLOS authors have the option to publish the peer review history of their article (what does this mean?). If published, this will include your full peer review and any attached files.

Reviewer #1: No

Reviewer #2: No

Reviewer #3: No

---

## [Author Response · Author response to Decision Letter 0]

21 Aug 2020

Our responses to each of the reviewers’ comments in a point-wise manner below.

In the revised submission, we have corrected the organization according to the formatting instructions provided by the journal.

2. Please amend either the abstract on the online submission form (via Edit Submission) or the abstract in the manuscript so that they are identical.

We have ensured that the Abstract submitted through the online submission form and that included in the revised article match.

3. We note that Figure 2-3 in your submission contain satellite images which may be copyrighted. All PLOS content is published under the Creative Commons Attribution License (CC BY 4.0), which means that the manuscript, images, and Supporting Information files will be freely available online, and any third party is permitted to access, download, copy, distribute, and use these materials in any way, even commercially, with proper attribution. For these reasons, we cannot publish previously copyrighted maps or satellite images created using proprietary data, such as Google software (Google Maps, Street View, and Earth). For more information, see our copyright guidelines: http://journals.plos.org/plosone/s/licenses-and-copyright.

Thank you for your concern regarding the copyright issues. However, we would like to assure you that the images included in this article do not contain any copyrighted satellite information. If you are referring to the colored features in images 2 and 3, they were created with open-source technology of the “Class activation mapping,” and a relevant reference has been cited in this regard. You can also find the information at the following website: https://github.com/zhoubolei/CAM

 

Reviewers' comments:

Reviewer #1: After initial screening I have observed following issues:

1. Authors should add more comparative analyses,

As recommended by the reviewer, we have added results of per lesion analysis including lesion detection rates and reading time of reviewers. We also found a correlation between the total number of extracted images and the number of images exceeding the significance threshold. The statistical correlations and interpretations have been discussed in the Results and Discussion sections.

2. It is advised to explain the equations present in the paper in more efficient manner.

We have explained the learning, validation, and test processes in AI development, and have provided more details about the parameters. We have also proved that random selection of images to construct each dataset was performed in an appropriate way by comparing the AUC values.

3. Some confusing and long lines are there, authors should adjust these lines.

We have ensured that there are no confusing sentences in the revised text. The manuscript has been professionally edited by an English language editing company and the editing certificate is being provided herewith for your perusal.

4. The literature can be improved by adding some recent plos one papers and also: Efficient Prediction of Drug-drug interaction using Deep Learning Models, Imputing missing values using cumulative linear regression, Visibility improvement and mass segmentation of mammogram images using quantile separated histogram equalisation with local contrast enhancement, Classification of COVID-19 patients from chest CT images using multi-objective differential evolution--based convolutional neural networks, Deep learning approach for microarray cancer data classification, Deep Transfer Learning based Classification Model for COVID-19 Disease.

We would like to thank the reviewer for the earnest suggestion. Unfortunately, we believe that the papers suggested for inclusion by the reviewer are not suitable for use as references in our article. However, PLoS ONE has published several valuable articles on the use of AI in medical imaging diagnosis and we have cited three of such papers in the revised manuscript.

5. Conclusion can be improved further by presenting more statistical computed values.

As recommended by the reviewer, we have enriched the information in the Discussion and Conclusions sections with statistical values.

Reviewer #2:

1. Author should reconsider the title. It is very confusing.

As suggested by the reviewer, we have modified the title for clarity. We have also got the manuscript edited by a professional English language editing company and have submitted the editing certificate.

2. Abstract is too long. Authors need to be rewrite it for better understanding.

We have shortened the abstract to 245 words. The re-written Abstract contains all the components prescribed in the author guidelines.

3. It is advised that ROC curve that is Figure 4 must be reevaluated by considering ROC analysis of competitive techniques.

We have explained the learning, validation, and test processes in AI development, and have provided more details about the parameters. We have also proved that random selection of images to construct each dataset was performed in an appropriate way by comparing the AUC values.

4. Figure 5 is confusing, so authors should provide more description about this figure.

We have added the results of per lesion analysis including lesion detection rates and reading time of reviewers. The statistical correlations and interpretations have been discussed in the Results and Discussion sections. In the revised manuscript, numbers that were previously not mentioned have been introduced. We are sure these additions will help readers in better understanding the graph.

5. It is advised to compare with more state-of-art techniques for comparative analyses.

Although our study is related to the recent research on AI, it mainly covers the field of clinical medicine. The primary objective of this study was to assess the clinical effectiveness of AI, not to prove the technical performance of AI. Analysis of whether the detection of clinical lesions could be improved with the help of AI was properly assessed using classical techniques of statistics. The results of the statistical analyses effectively demonstrated the differences between the two reviewer groups. In addition, the AI technology used to assist the CE reading was also selected from the recently developed high-performance networks. As reported in the reference number 11 cited in the manuscript, the Inception-ResNet-V2 shows the third highest accuracy among various deep learning models. Because the other two models with high accuracy involve many more parameters, we selected Inception-ResNet-V2 to evaluate the accuracy and training/inference time simultaneously.

6. Parameter setting of the proposed model should be presented in tabular form.

We have presented the details of the parameters in Table 1 in the revised manuscript as suggested by the reviewer.

7. There are some typos and grammatical mistakes present in the manuscript.

We have corrected all the typographical and grammatical errors. The editing certificate is being submitted with the revised manuscript.

8. Paper organization is missing at the end of introduction

We have organized the manuscript in accordance with the formatting instructions provided by the journal.

9. Headings and sub-headings have not been marked

In the revised paper, we have marked the headings and sub-headings according to the formatting instructions.

10. It is advisable to authors consider following useful references

Fusion of medical images using deep belief networks, Classification of COVID-19 patients from chest CT images using multi-objective differential evolution–based convolutional neural networks, Efficient prediction of drug–drug interaction using deep learning models, Deep Transfer Learning based Classification Model for COVID-19 Disease, Multi-objective differential evolution based random forest for e-health applications

Unfortunately, the papers suggested by the reviewer are not suitable for use as references in our article. However, PLoS ONE has published several valuable papers on the use of AI in medical imaging diagnosis and three of such papers have been cited in the revised manuscript.

Reviewer #3: Paper can be improved further

1. Add suitable research motivation

We have modified the Introduction section to highlight the motivation for this research. We have emphasized upon the distinction of our study and have stated the goals clearly. We have presented a practical comparison method considering the reality in capsule endoscope reading and have stated that achieving a statistical interpretation through the use of the presented method was the goal of our study.

2. Add more relevant references from plos one.

PLoS ONE has published several valuable papers on the use of AI in medical imaging diagnosis and three of such papers have been cited in the revised manuscript.

3. Equations should be explained carefully

We have explained the learning, validation, and test processes in AI development, and have provided more details about the parameters. We also proved that random selection of images to construct each dataset was performed in an appropriate way by comparing the AUC values.

4. More comparative analysis are required

As suggested by you, we have added results of the per lesion including lesion detection rates and reading time of reviewers. We also found a correlation between the total number of extracted images and the number of images exceeding the significance threshold. The statistical correlations and interpretations have been discussed in the Results and Discussion sections.

5. Add statistical analysis in conclusion or discussion session

As suggested by the reviewer, we have enriched the Discussion and Conclusions sections by including the results of statistical analysis.

---

## [Decision Letter · Decision Letter 1]

16 Oct 2020

Artificial intelligence that determines the clinical significance of capsule endoscopy images can increase the efficiency of reading

PONE-D-20-15923R1

Dear Dr. LIM,

We’re pleased to inform you that your manuscript has been judged scientifically suitable for publication and will be formally accepted for publication once it meets all outstanding technical requirements.

Kind regards,

Sudipta Roy

Academic Editor

PLOS ONE

Additional Editor Comments (optional):

The paper is technically sound and did in-depth analysis. Authors have address all the points very carefully.

Reviewers' comments:

Reviewer's Responses to Questions

**Comments to the Author**

1. If the authors have adequately addressed your comments raised in a previous round of review and you feel that this manuscript is now acceptable for publication, you may indicate that here to bypass the “Comments to the Author” section, enter your conflict of interest statement in the “Confidential to Editor” section, and submit your "Accept" recommendation.

Reviewer #3: All comments have been addressed

2. Is the manuscript technically sound, and do the data support the conclusions?

Reviewer #3: Yes

3. Has the statistical analysis been performed appropriately and rigorously? 

Reviewer #3: Yes

4. Have the authors made all data underlying the findings in their manuscript fully available?

Reviewer #3: Yes

5. Is the manuscript presented in an intelligible fashion and written in standard English?

Reviewer #3: Yes

6. Review Comments to the Author

Reviewer #3: (No Response)

7. PLOS authors have the option to publish the peer review history of their article (what does this mean?). If published, this will include your full peer review and any attached files.

Reviewer #3: No

---

## [Editor Report · Acceptance letter]

20 Oct 2020

PONE-D-20-15923R1 

Artificial intelligence that determines the clinical significance of capsule endoscopy images can increase the efficiency of reading 

Dear Dr. LIM:

I'm pleased to inform you that your manuscript has been deemed suitable for publication in PLOS ONE. Congratulations! Your manuscript is now with our production department. 

Kind regards, 

on behalf of

Dr. Sudipta Roy 

Academic Editor

PLOS ONE